# Reproducibility study of "Robust Fair Clustering: A Novel Fairness Attack and Defense Framework"

**Lucas Ponticelli**                                                  *lucas.ponticelli@student.uva.nl*
*University of Amsterdam, Amsterdam, the Netherlands*

**Vincent Loos**                                                                 *vincent@vloos.net*
*University of Amsterdam, Amsterdam, the Netherlands*

**Eren Kocadag**                                                      *eren.kocadag@student.uva.nl*
*University of Amsterdam, Amsterdam, the Netherlands*

**Kacper Bartosik**                                                 *kacper.bartosik@outlook.com*
*University of Amsterdam, Amsterdam, the Netherlands*

*https://openreview.net/forum?id=Xu1sEPhjqH*

## Abstract

This reproducibility study examines "Robust Fair Clustering: A Novel Fairness Attack and Defense Framework" by Chhabra et al. (2023), an innovative work in fair clustering algorithms. Our study focuses on validating the original paper's claims concerning the susceptibility of state-of-the-art fair clustering models to adversarial attacks and the efficacy of the proposed Consensus Fair Clustering (CFC) defence mechanism. We employ a similar experimental framework but extend our investigations by using additional datasets. Our findings confirm the original paper's claims, reinforcing the vulnerability of fair clustering models to adversarial attacks and the robustness of the CFC mechanism.

## 1   Introduction

Clustering algorithms can be used in different contexts that have consequences on the lives of people in societies (Ghosal et al., 2020). In the past few years, clustering algorithms have been developed to ensure that performing tasks such as awarding home loans or predicting recidivism are done in a way that is fair and unbiased. In recent times, there has been a notable number of works in this research domain (Bera et al., 2019).

"Robust Fair Clustering: A Novel Fairness Attack and Defense Framework" by Chhabra et al. (2023) portrays how susceptible existing fair clustering algorithms are to adversarial attacks. The authors also introduce a novel black-box attack, which decreases the fairness performances of clustering models. Moreover, the Consensus Fair Clustering (CFC) algorithm is introduced, which is resilient to these attacks.

In this paper, we aim to reproduce the work done by Chhabra et al. (2023) and build upon it by performing similar experiments in new contexts. The following contributions are made:

- All of the work done by Chhabra et al. is reproduced by experiments that we conducted. Through this, the claims made in the original paper are supported.

- We extend upon the work of the original authors by performing the experiments on datasets with different characteristics than the ones utilized in the original study. By doing this, the robustness of the novel fair clustering algorithm is evaluated.

## 2 Scope of reproducibility

Chhabra et al. (2023) address the vulnerability of fair clustering algorithms to adversarial attacks and propose a defence mechanism for these attacks. The authors make the following three claims:

- **Claim 1:** State-of-the-art fair clustering models are highly susceptible to adversarial attacks, which can significantly diminish their fairness performance.

- **Claim 2:** The novel black-box attack introduced can effectively degrade the fairness performance by altering a small portion of protected group memberships.

- **Claim 3:** The proposed Consensus Fair Clustering (CFC) defence mechanism not only resists adversarial attacks but can also maintain or even improve clustering performance post-attack.

The authors' work expands the concept of fairness in machine learning by introducing a novel approach: ensuring fairness while faced with adversarial challenges. It emphasises the importance of developing AI systems that can maintain fairness when under attack, expanding the conversation beyond static criteria of algorithmic fairness. This research also opens up chances to investigate and compare different strategies for fair clustering under adversarial conditions, contributing to the broader discussion on building fair AI systems.

## 3 Methodology

This section delves into the specifics of our approach to reproduce the paper by Chhabra et al. (2023). It describes the used algorithms, the different datasets, the utilised hyper-parameters, the experimental setup of our study, the explanation of the different metrics and finally the computational requirements and environmental impact.

### 3.1 Models description

In their study, Chhabra et al. (2023) introduce three key clustering algorithms: Fair K-Center (KFC) (Harb & Lam, 2020), Fair Spectral Clustering (FSC) (Kleindessner et al., 2019), and Scalable Fairlet Decomposition (SFD) (Backurs et al., 2019), all designed for fairness-aware clustering.

The KFC algorithm is tailored for fairness in clustering, particularly within the realm of the k-center clustering problems where the maximum distance between any data point and its assigned cluster center is minimized. FSC, a fairness-aware clustering algorithm, concentrates on spectral clustering, a technique utilising the eigenvectors of a similarity matrix for clustering. SFD, with a specific focus on fairness in clustering, employs fairlet decomposition: a method for partitioning data into small, balanced clusters. In essence, these algorithms tackle fairness concerns in clustering by integrating fairness-aware objectives or constraints into their methodologies.

Chhabra et al. (2023) attack these clustering algorithms using their novel black-box attack algorithm. This algorithm aims to reduce the fairness utility of the fair clustering algorithms by perturbing a small percentage of samples' protected group memberships. The problem can be summarised as a minimisation problem where a given measure of fairness utility is the objective:

$$\min_{G_A} \varphi(\theta(O, G_D), G_D) \text{ s.t. } O = F(X, K, \eta(G_A, G_D)).$$

**Where:** $G_A$ is a small portion of the protected groups, $G_D$ is the entire set of protected groups, F is a fair clustering algorithm, K is the number of clusters, X is the dataset, $\eta$ and $\theta$ are mapping functions and $\varphi$ is a given fairness utility function. This utility function is trying to find the best configuration of clusters ($K$) such that the clustering outcome is fair with respect to the entire set of protected groups ($G_D$), even when a small portion (chosen randomly) of these groups ($G_A$) is deliberately manipulated by an adversary. The

fairness utility function evaluates how fair these clusters are, and the goal is to minimize this evaluation, ensuring fairness in the clustering outcome despite potential adversarial interference.

The above problem is a two-level hierarchical optimisation problem (Anandalingam & Friesz, 1992). To obtain solutions for this problem, RACOS (Yu et al., 2016) is used due to its known theoretical guarantees and the non-convex and non-differentiable nature of the problem.

For the defence part, Consensus Fair Clustering (CFC) is introduced, as depicted in Chhabra et al. (2023). The method efficiently transforms the consensus clustering problem into a graph partitioning task. It leverages a novel graph-based neural network architecture for learning representations tailored to fair clustering. CFC adeptly addresses data and algorithmic challenges posed by attacks through a concise two-stage process.

In the initial stage, a subset of training data is selectively sampled, and cluster analysis is conducted to derive the basic partition and co-association matrix. Given that poisoned samples constitute a minor fraction of the overall training data, their probability of inclusion in the subset is minimised, mitigating their negative impact. In the second stage, CFC integrates the basic partitions with a fairness constraint, thereby enhancing the algorithmic robustness of the approach.

For a full explanation of the attack and defence mechanisms, we refer to the paper by Chhabra et al. (2023). The paper has an excellent illustration showing the two stages intuitively, shown in Figure 5 adopted from Chhabra et al. (2023).

### 3.2 Datasets

In this study, we utilised various datasets, specifically Extended-Yale-B(Vishwakarma & Dalal, 2020), MNIST-USPS (Hull, 1994; LeCun et al., 2010), Office-31(Herath et al., 2016), and Digits (Alpaydin & Kaynak, 1998), which were included in the original study. These datasets were acquired following a methodology similar to that outlined by the authors. Furthermore, we employed the Dutch census, the Open University Learning Analytics dataset (OULAD) and a subset of the FairFace dataset (Karkkainen & Joo, 2021). The statistics for all datasets are presented in Table 1.

The Dutch census and OULAD originate from a collection of datasets presented in *"A survey on datasets for fairness-aware machine learning"* (Le Quy et al., 2022). These datasets are both tabular, allowing us to investigate how the models handle a different data structure than the image datasets in the original research. On the other hand, FairFace is another image dataset with the key difference being that the protected attribute, sex, is a better real-life example compared to the arbitrary chosen features in the original paper. Due to the large size of this dataset and our limited computational capacity, we only included the samples from the validation set where the race is black or white.

In Table 1 the protected attribute column is the focus of our investigation. The *Source* attribute concerns which sub-dataset an example belongs to. *Normal-colour inverted* focuses on whether the example is a normal image or if it is colour inverted. Finally, the Lighting-Elevation concerns the differences in lighting and elevation of each face between images.

This research centres around unsupervised clustering problems, rendering the division of datasets into train, validation, and test splits inapplicable.

The preprocessing of the datasets from the original paper was already taken care of by the authors. The datasets we used to extend the authors' research were preprocessed in the following way. As a basis, the provided code from the authors was used, and the features were adapted to work with their code, for example selecting the categorical features or the protected feature.

Instructions to download the datasets are included in our GitHub repository[1].

---

[1]Code available at: `https://github.com/vinloo/robust-fair-clustering-reproducibility`

| Dataset | Sub-Dataset | Samples Size | Size Images \| Num. feature | Categories Count | Average entries per category | Protected attribute |
|---|---|---|---|---|---|---|
| Extended Yale B | n/a | 2414 | 192x160 | 38 | 64 | Lighting-Elevation |
| MNIST-USPS | MNIST | 70000 | 28x28 | 10 | 7000 | Source |
| | USPS | 9298 | 16x16 | 10 | 920 | |
| Office-31 | DSLR | 498 | 4288x2818 | 31 | 15 | Source |
| | Webcam | 795 | 640x480 | 31 | 20 | |
| Digits | n/a | 1797 | 8x8 | 10 | 180 | Normal-Color inverted |
| Dutch census | n/a | 60420 | 12 | 2 | 30210 | Sex |
| FairFace | n/a | 3641 | 224x224 | 2 | 1820 | Sex |
| OULAD | n/a | 21562 | 12 | 2 | 10281 | Sex |

Table 1: Statistics for utilised datasets.

### 3.3 Hyper-parameters

The algorithms and hyper-parameters utilised in this study were following the specifications outlined in the main paper (Chhabra et al., 2023). We adopted the default hyper-parameters provided in the original implementations, aligning with the choices made by the authors. Table 3 in Appendix C shows an overview of the used hyper-parameters for the deployed algorithms.

We utilised grid search to find the best *alpha* and *beta* values for each of our datasets. For the *budget* hyper-parameter, we chose to adopt the average of the values used in the original study to maintain consistency with the experimental setup, ensuring fair comparisons across datasets and enabling validation of algorithm performance under similar optimization settings. The *name_bal* variable in ConsensusFairClustering (CFC) was also set to 0.1 as a threshold as the dataset domains we are dealing with are as *sensitive* as the Yale dataset when it comes to *fairness balance*.

A brief explanation of the hyper-parameters in Table 3 (Appendix C):
*Beta*: Denotes the approximation guarantee of the k-median algorithm (Backurs et al., 2019).
*Alpha*: Size of a probability distribution (Backurs et al., 2019).
*Delta*: Parameter used to control different parameters inside the algorithm (Harb & Lam, 2020).
*Num. of Neighbours*: Number of neighbouring points used to calculate the cluster assignment.
*Metric*: A specific formulation to calculate the distance, can influence the way an algorithm assigns clusters.
*Budget*: Represents the count of solution evaluations the optimiser will perform, which is directly linked to the duration of the optimisation process.
*Num. of Clusters*: Number of clusters the algorithm generates.
*Random-State*: The different seeds ensure that the results are not conditioned on a specific initialisation.

### 3.4 Experimental setup and code

To replicate the results of the study we ran the attack on SFD, FSC and KFC, and the defence, with the novel CFC algorithm, on the four datasets mentioned in Section 1. As done in the original paper, we repeat all experiments ten times with a different random seed and report the average and standard deviation of the metrics described in Section 3.5. The seeds we used to replicate the experiments were the same ones chosen in the original paper.

Furthermore, where the authors originally attack primarily the balance metric, and only attack entropy for a given algorithm-dataset combination when the balance starts at zero, we attack both balance and entropy for all combinations to verify the claim that balance is a more robust measure of fairness than entropy. By simultaneously targeting and measuring both balance and entropy, we can confirm the effectiveness of their attacks.

To expand upon the authors' study, we further implemented two attack algorithms and the defence algorithm across two tabular datasets and one image dataset. The extension of the research by Chhabra et al. (2023) was facilitated by the integration of a basic preprocessing codebase for tabular datasets. This addition facilitated the use of new datasets in their existing code and allowed for the execution of further experiments.

The source code and documentation necessary to replicate our research findings are available in our dedicated GitHub repository[1]. The challenges encountered during our attempt to replicate the findings and extend the research of the paper are detailed in Section 5.2.

### 3.5 Measure descriptions

In our study, emphasis is placed on the evaluation of all the metrics from Chhabra et al. (2023) namely, normalised mutual information (NMI), Accuracy (ACC), Balance and Entropy. For fully detailed definitions we refer to appendix B of Chhabra et al. (2023). To summarise, NMI is essentially a normalised version of the widely used mutual information metric and ACC is an unsupervised equivalent of the traditional classification accuracy. Balance and entropy are both measures of fairness, where a higher score indicates more fairness.

### 3.6 Computational requirements

In the conducted experiments, we employed the supercomputer sourced from the Dutch National Supercomputer Snellius[2]. We utilised a single NVIDIA A100 GPU and 18 cores of an Intel Xeon Platinum 8360Y processor to train our models requiring GPU computing. For our models requiring CPU computing, we used 32 cores from 4 AMD EPYC 7F32. To carry out initial experiments and debug the code-base, we utilised local machines with varying specifications and negligible energy consumption. All together the experiments took approximately 270 hours to run. The computational time it took for each experiment can be found in Table 2.

| Algorithms | MNIST-USPS | Office-31 | Extended Yale B | Digits | OULAD | Dutch census | FairFace |
|---|---|---|---|---|---|---|---|
| **SFD** | 05:38:13 | 03:02:35 | 06:09:00 | 01:22:07 | 15:37:52 | 57:20:33 | n/a |
| **FSC** | 06:07:14 (on GPU) | 42:13:01 | 10:56:38 (on GPU) | 03:36:44 (on GPU) | n/a | n/a | n/a |
| **KFC** | 00:49:01 | 08:49:24 | 15:02:01 | 00:41:52 | 00:58:50 | 02:56:02 | 09:19:24 |
| **CFC** | 04:41:45 (on GPU) | 02:44:46 (on GPU) | 11:16:46 (on GPU) | 02:31:59 (on GPU) | n/a | n/a | 61:43:25 (on GPU) |

Table 2: Computational time to reproduce each experiment (hh:mm:ss)

Utilising the online tool available at `https://mlco2.github.io/impact/#compute`, we estimated the carbon emissions to amount to approximately 10.02 kg CO2 equivalent through 170 hours of CPU computing, assuming a carbon efficiency of 0.421 kg/kWh, according to `https://www.nowtricity.com/country/netherlands/` for the year 2023 in the Netherlands. Additionally, the GPU computing activities resulted in an emission of around 11.33 kg CO2 equivalent over 100 hours. In total, this approximates driving around 85 km with an average internal combustion engine car[3]. The carbon emission figures are estimated because we were unable to obtain precise measurements of emissions for Snellius when using its partitions, the actual emissions should be lower than what is estimated here.

## 4 Results

To verify the results from the original study, this section presents our reproduction results. Additionally, it contains the outcomes of further experiments conducted beyond the scope of the original paper. The results of our experiments align with the results achieved by Chhabra et al.

---

[2]We thank SURF (`www.surf.nl`) for the support in using the National Supercomputer Snellius.

[3]`https://www.epa.gov/energy/greenhouse-gases-equivalencies-calculator-calculations-and-references`

### 4.1 Results reproducing original paper

Two separate experiments evaluate the claims made by the original authors. These claims appear in Section 2. The experiments assessing the attack address claims 1 and 2, while those evaluating the defence focus on claim 3.

As in the original paper, we performed the attack on all four datasets that the authors used in the original paper, namely MNIST-USPS, Office-31, Yale and DIGITS datasets. Our reproduction results with the FSC, KFC and SFD algorithms are visible in Figures 1, 2 and 3 respectively. The graphs from the original paper that our results are compared to, can be found in Figures 6 and 7 of appendix D adopted from the original paper (Chhabra et al., 2023). The results of the FSC attack show similar trends in our results compared to the results from the original paper. However, in some attacks on the metrics, the ranges and trends

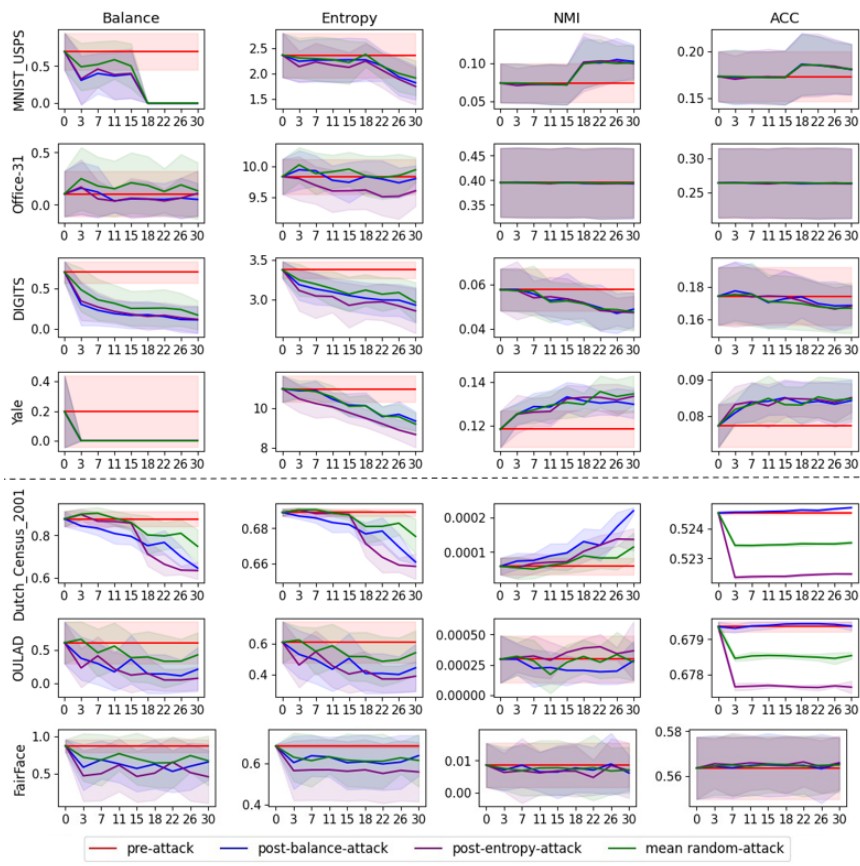

Figure 2: KFC results for all datasets (x-axis: % of samples attacker can poison).

of the values that the scores take differ from those in the original paper. An example is the post-entropy attack on the MNIST-USPS and DIGITS datasets, where our post-entropy values are larger than the original values. For the KFC and SFD attacks, the trends and range of all metrics are close to the original on the MNIST-USPS, DIGITS and Yale datasets. On the Office-31 dataset, the trends of the post-balance and post-entropy scores mostly align, with only moderate differences observed. The original authors claimed that State-of-the-art fair clustering models are highly susceptible to adversarial attacks and that their novel black-box attack can effectively degrade the fairness performance. In our results, we generally see that as more samples can be perturbed by an adversary, the post-attack fairness decreases. This is especially visible in the KFC and SFD attacks. In the FSC result, this is less clear. One thing that can also be seen is that

the pre-attack values are higher than the post-attack fairness scores. Observing this, one could conclude that claims 1 and 2 made by the original authors as described in Section 2 hold.

Figure 4 presents the results of the experiments using their novel CFC defence mechanism. The third claim of the original work states that CFC not only resists adversarial attacks but can also maintain or even improve clustering performance post-attack. Looking at the graphs in Figure 4 and the last two rows of Figure 7, the displayed trends seem somewhat different in our results in comparison to those of the original authors. However, the results found are in line with their standard deviations, as described by Chhabra et al. (2023), except for the balance metric on the Yale dataset. Despite this, it could be argued that the defence adequately resists adversarial attacks in our results, as the fairness metrics and clustering performance did not get worse in comparison to the pre-attack when the number of samples poisoned by the adversary increased. Once more, the only exception here could be the performance metric on the Yale dataset, which does return lower NMI and accuracy scores when a higher percentage of samples are perturbed by the attacker.
At times it can be seen that the clustering performance is higher post-attack in comparison to pre-attack. An example of this is the results for the NMI and ACC metrics on the MNIST-USPS dataset. Considering that CFC resists the adversarial attacks, and maintains clustering performance, one could conclude that claim 3 holds.

### 4.2 Results beyond original paper

Below each dotted line in Figures 2 and 3, the results of the attacks beyond the scope of the original paper are shown. For the attack on the KFC algorithm, it is clear that the post-attack fairness scores are worse than the pre-attack scores. However, this is less obvious in the SFD algorithm. For the attack on the SFD algorithm using the Dutch Census dataset, the attacks do not decrease in fairness scores (balance and entropy) post-attack. On the OULAD dataset, the fairness scores did become worse with the SFD algorithm as well. This discrepancy suggests that the constancy of fairness scores in the Dutch Census dataset may be less about the resiliency of the attack method and more indicative of the clustering model's unsuitability for the dataset in question. NMI scores, which are notably low, imply that the clustering output is nearly random, thereby questioning the model's effectiveness on each dataset. Consequently, these results imply that the clustering model's limitations on the Dutch Census dataset are a significant factor, and it would be misleading to ascribe the observed robustness in fairness scores solely to the efficacy of the attack techniques.

As seen in the last row of Figure 4, the defence mechanism of the original authors was also tested on a new dataset (the FairFace dataset). The pre-attack line is invisible in these subplots as it is covered by the line of the post-attack metric that is attacked. This happens because both lines have the same values for each percentage of samples that the attacker can poison. Therefore, the defence does resist adversarial attacks and maintains clustering performance post-attack on this data. Considering these observations, claim 3 as described in Section 2 does hold on this dataset that the original authors did not work with.

Finally, we investigated whether balance is a more robust metric of fairness than entropy, and if it also results in a larger decrease in fairness when it is the metric attacked. Looking at Figures 1, 2, 3, and 4, a post-balance-attack and post-entropy-attack line are present, highlighting the scores after attacking balance and entropy respectively. Generally, the attacked metric showed a lower score compared to when the other metric was targeted. However, there were instances where the balance, theoretically more robust, actually decreased more when entropy was attacked. This suggests that despite the balance of theoretical robustness, it might not always be the most effective target in practical scenarios. Thus for attackers, targeting a variety of fairness metrics might be the most advantageous approach.

## 5 Discussion

Overall, we were able to reproduce the results that the authors found and verify that their three claims indeed hold in different contexts with various datasets. Thanks to the presence of seed numbers for the balance and entropy attacks on the four provided datasets, we were able to find similar results as the authors. However, due to limits in our resources, we were not able to run the novel CFC defence method several times to obtain a standard deviation and results that were more aligned with theirs. The lack of seed numbers on their

different runs for their defence algorithm made it impossible for us to replicate their exact results which explains the slight differences in our results when trying to reproduce their experiment. The third claim about their innovative defence mechanism holds to a certain degree. We observed that with a new dataset, their defence maintains the same level of fairness and performance as before an attack, but it doesn't enhance them. To fully validate this claim, it would be beneficial to test their algorithm on additional datasets, using various seeds and conducting multiple trials.

This last section of our research highlights the aspects of the authors' research that were straightforward to reproduce, alongside those that posed significant challenges. Through the next subsections, we will detail which parts of the reproduction were easier and which caused us significant challenges. In the last subsection, we document our interactions with the original authors.

## 5.1   What was easy

The authors already provided a repository with a codebase to reproduce the results. As we will discuss in Section 5.2, this codebase was far from optimal and user-friendly but provided a solid foundation to run the experiments. Moreover, the codebase included code to run the experiments on tabular datasets although this was not utilised in the original paper and did not work with the defence algorithm. This allowed us to extend the research to different datasets with a tabular structure for the attack experiments.

## 5.2   What was hard

Although the authors have provided a repository with code to reproduce their results, some difficulties were met when trying to execute it. Most importantly, the code was completely devoid of documentation, which made it challenging to solve the issues we faced during the replication. Moreover, even though a Python module requirements file was provided, it was incomplete and did not contain specific module versions which caused the code to malfunction. We had to make some changes to the original code base to make it work properly. This was a rather time-consuming process and could have easily been avoided if the code had been documented properly. In Appendix B we provide an overview with all changes we had to make to the original code. This does not include our code for additional experiments.

Besides the issues regarding the codebase, we also faced several dataset-related challenges. Firstly, the URL to the Yale dataset included in the code, which automatically downloads the dataset, was not responsive. Therefore, we had to manually download the dataset from a mirror and update the code accordingly. The fact that the original dataset could not be used is likely the cause for the result differences witnessed in section 4.1. Despite this, the results were comparable, which suggests that the difference in the version of the Yale dataset had no significant impact on the conclusions addressed in section 5. Moreover, to use a given dataset with the provided code, several Numpy arrays had to be created specifically for the dataset. Fortunately, these arrays were already included in the repository for the four base datasets. However, creating these arrays for the additional datasets was not straightforward at all as they did not have an intuitive definition, were not discussed in the paper, were not documented, and no code was provided to create them for the base datasets. This meant we had to deeply analyse the vectors and the code to gain an understanding of what purpose they served.

Finally, all code for the attack experiments only worked on CPU. For the SFD and KFC algorithms, this was not a serious issue as those algorithms were fast enough on CPU. However, the FSC algorithm had an expected runtime of several days for a single dataset, which was unfeasible to run within our time constraints. The original code for the FSC algorithm contained a k-means implementation which did not support GPU-acceleration. We simply changed the code to use an alternative module which allows GPU-acceleration to reduce the total runtime to several hours. Due to limited resources, we were unable to run FSC on our two new tabular datasets as debugging the code for these datasets proved too resource-intensive after their inclusion into the experiment.

## 5.3 Communication with original authors

We have contacted the original authors to improve our understanding of the data and the code provided. The main author responded and answered a subset of our questions and delegated the unanswered questions to one of their co-authors. A week later, we got a response from this author, who provided us with answers to the previously unanswered questions.

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

## A  Proposed CFC framework by Chhabra et al. (2023)

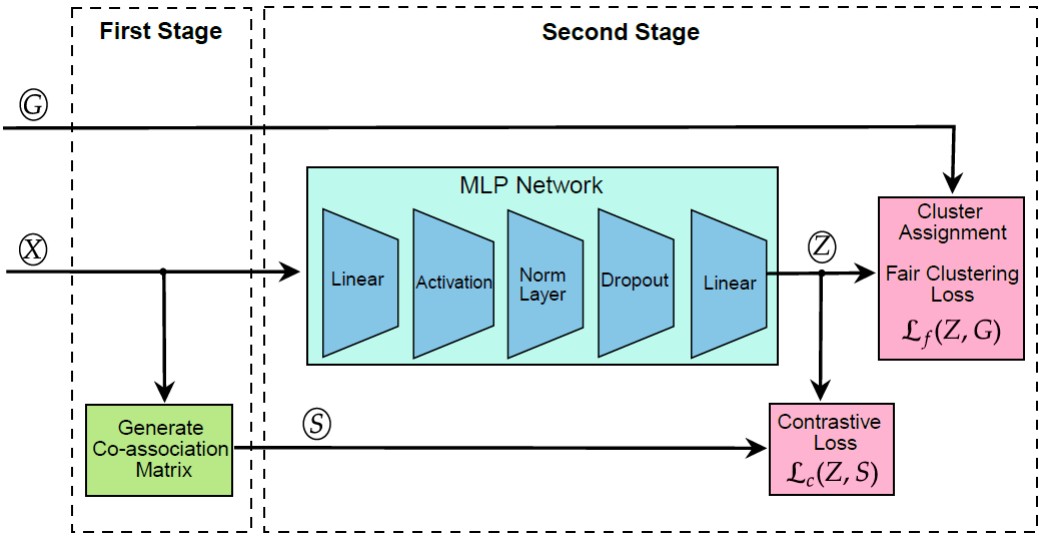

Figure 5: Proposed CFC framework. Adopted from Chhabra et al. (2023).

## B  Modifications to the original codebase

| File | Reason | Change |
|---|---|---|
| requirements | Missing modules and incorrect versions | Added `pandas`, `seaborn`, `networkx`, `chardet`, `pyckmeans`, `kmeans-pytorch` and changed `scikit-learn` to version `0.22.2.post1` |
| `mnist_usps.py`, `extended_yaleB.py`, `office31.py` | `FileNotFoundError` raised when loading the MNIST_USPS, Yale or Office-31 dataset due to use of absolute paths | Removed the path prefix from `dataset_dir` on line 13 |
| `base.py` | `TypeError` being raised when loading data, possibly due to version mismatch | Changed `np.int` to `np.int8` on lines 75 and 76 |
| `extended_yaleB.py` | Dead URL for the MNIST_USPS dataset, only a manual download was available | Removed the automatic download code |
| `fair_kcenter.py` | The exception was raised when "IBM's CPLEX is not installed" is supressed. Now a user never knows why KFC does not work or even that CPLEX is required to run it | Removed the try-except block supressing the exception |
| `fair_spectral.py` | The CPU-only version of k-means is too slow | Changed the k-means module to `kmeans-pytorch` |

# C  Overview employed Hyper-parameters

| Algorithm | Dataset | Beta | Alpha | Delta | Num. of Neighbours | Metric | Budget | Num. of Clusters | Random-State |
|---|---|---|---|---|---|---|---|---|---|
| KFC | MNIST-USPS | n/a | n/a | 0.1 | n/a | n/a | 30 | Number Unique labels | 150, 1, 4200, 424242, 1947, 355, 256, 7500, 99999, 18 |
| | Office-31 | | | | | | 20 | | |
| | Digits | | | | | | 15 | | |
| | Extended Yale B | | | | | | 20 | | |
| FSC | MNIST-USPS | n/a | n/a | n/a | 3 | Euclidean | 10 | | |
| | Office-31 | | | | | Manhattan | 20 | | |
| | Extended Yale B | | | | | | 10 | | |
| | Digits | | | | | | 15 | | |
| SFD | MNIST-USPS | 2 | 5 | n/a | n/a | n/a | 50 | | |
| | Office-31 | | | | | | 20 | | |
| | Extended Yale B | | | | | | 20 | | |
| | Digits | 1 | | | | | 25 | | |
| CFC | MNIST-USPS | 25 | 100 | n/a | n/a | n/a | n/a | | |
| | Office-31 | 100 | 1 | | | | | | |
| | Extended Yale B | 10 | 50 | | | | | | |
| | Digits | 50 | 10 | | | | | | |

Table 3: Overview of the utilised hyper-parameters

# D   Original attack results

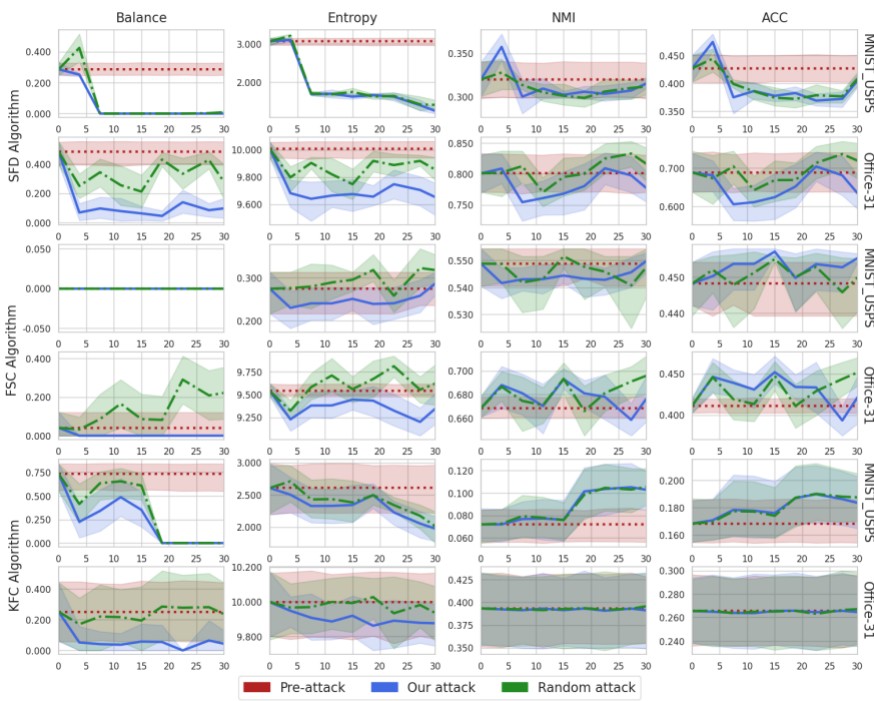

Figure 6: Attack results on MNIST-USPS and Office-31 (x-axis: % of samples attacker can poison). Adopted from Chhabra et al. (2023).

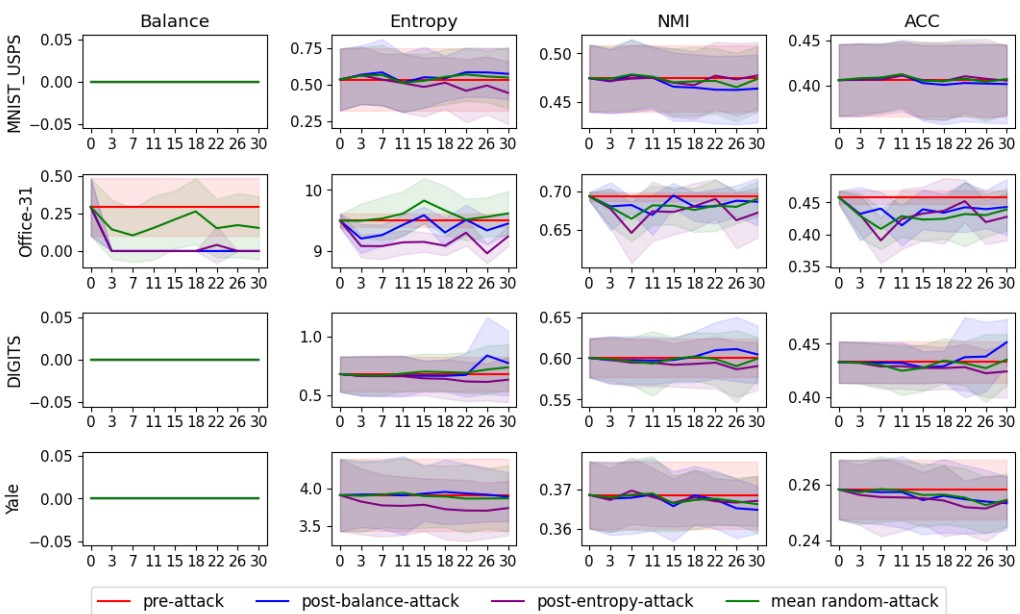

Figure 1: FSC results for all datasets (x-axis: % of samples attacker can poison).

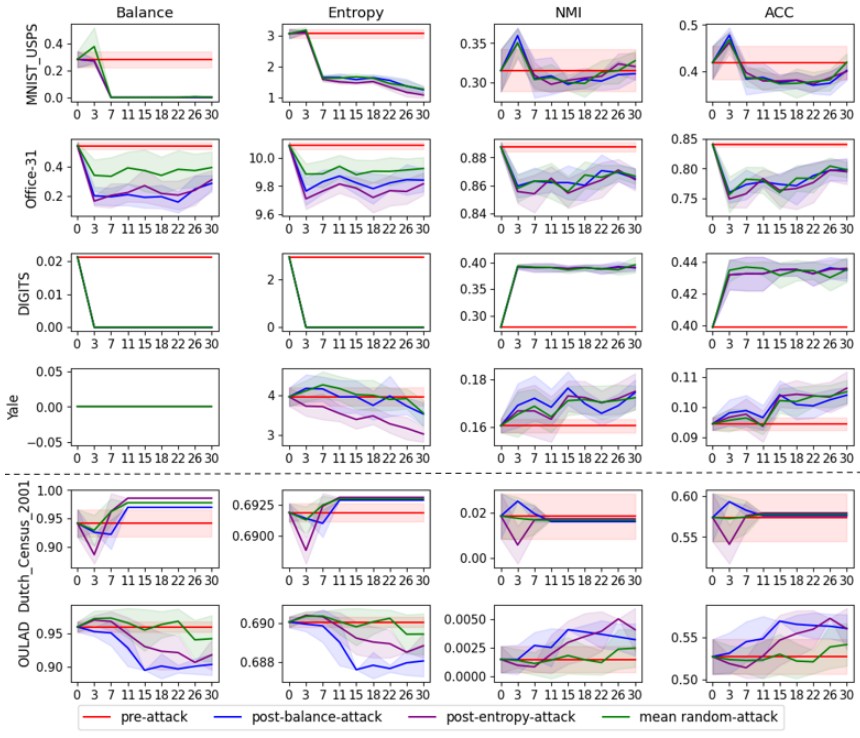

Figure 3: SFD results for all datasets (x-axis: % of samples attacker can poison).

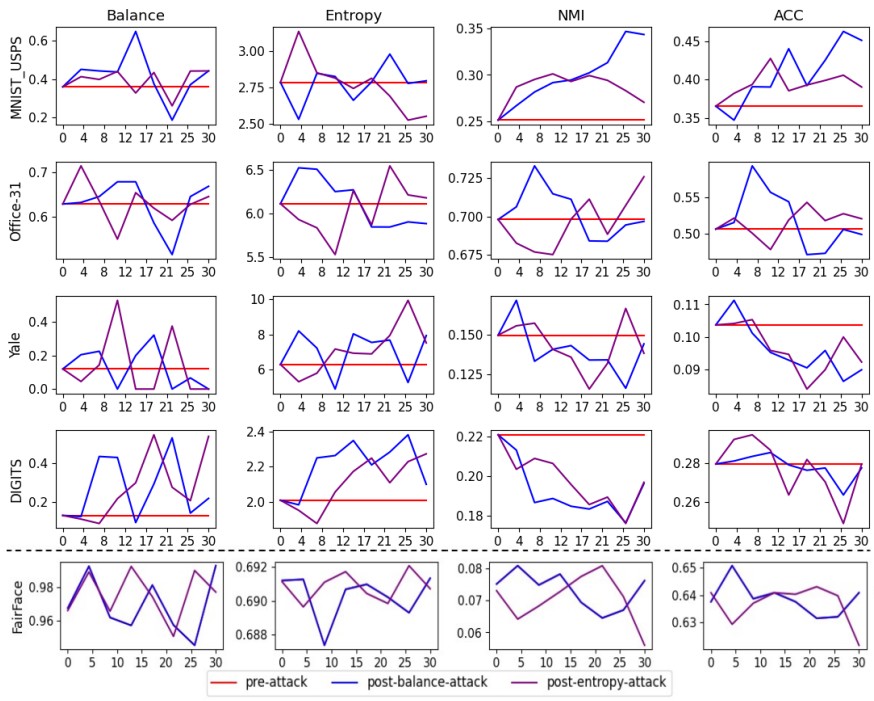

Figure 4: CFC results for all datasets (x-axis: % of samples attacker can poison).

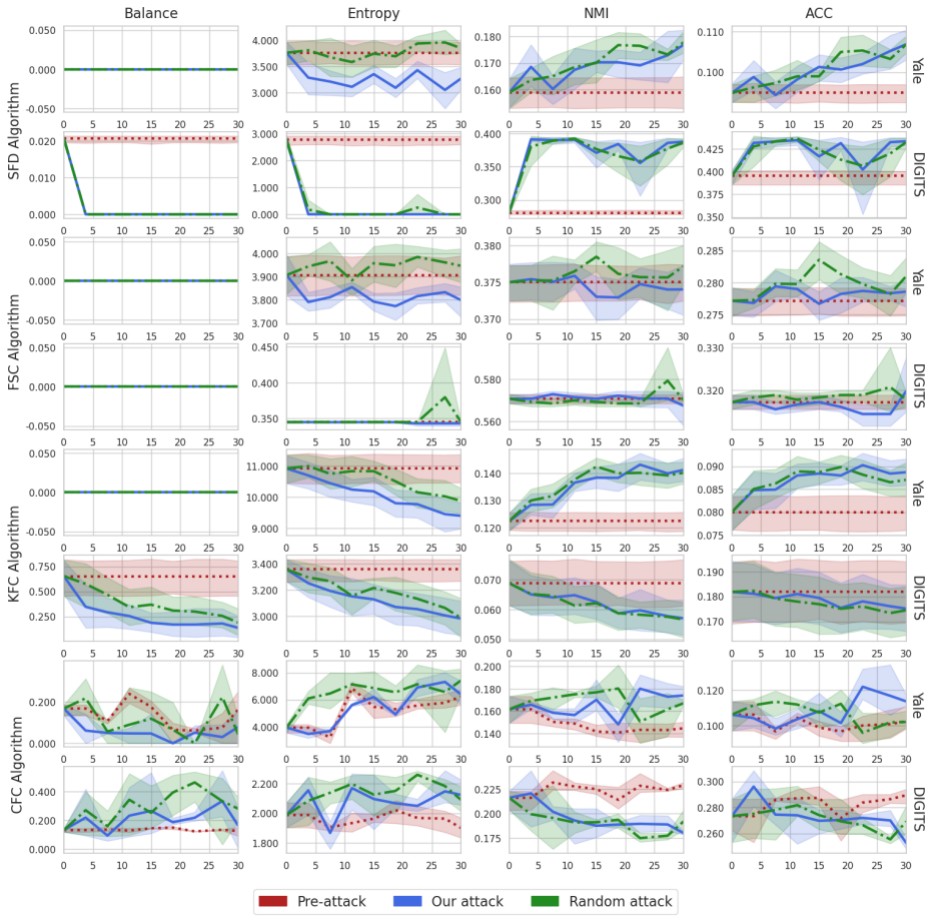

Figure 7: Attack results Yale and DIGITS (x-axis: % of samples attacker can poison). Adopted from Chhabra et al. (2023).

