# OpenReview forum: "Reproducibility study of "Robust Fair Clustering: A Novel Fairness Attack and Defense Framework""
_TMLR — Accepted by TMLR_

### Review · Reviewer_5aS3 · 2024-03-03

**Summary Of Contributions:**

The paper is a reproducibility study of the ICLR'23 paper by Chhabra et al titled "Robust Fair Clustering: A Novel Fairness Attack and Defense Framework". The authors undertake experiments to validate and confirm the original paper's findings such as (1) the vulnerability of fair clustering models to adversarial attacks and (2) the adversarial robustness of the CFC fair clustering model. The paper also considers additional datasets for experiments not used in the original work, such as Dutch Census, OULAD, and FairFace.

**Audience:**

Yes

**Broader Impact Concerns:**

N/A.

**Claims And Evidence:**

No

**Requested Changes:**

I believe reproducibility studies are very important and that this work conducts an extensive set of experiments to validate Chhabra et al's work. There are a few issues regarding unsubstantiated claims and conclusions, formatting, and typos, which I have outlined in the Weaknesses section above. If the authors make the aforementioned requested changes, I am happy to recommend acceptance.

**Strengths And Weaknesses:**

**Strengths**:
- The experiments to reproduce the work of Chhabra et al consitute a methodologically correct approach that follows the original paper closely. The results obtained also follow similar trends and affirm the findings of the original paper, which is a positive sign.
- The authors aim to extend the original approach to 3 additional datasets beyond the original paper (there are some issues here-- please see below).
- The authors investigate whether attacking balance or entropy is more advantageous for the attacker.

**Weaknesses**:
1. In the abstract the authors state: *Furthermore, our study adds to the field of fair clustering algorithms by identifying challenges and offering insights, guiding future efforts in the development of more robust and equitable AI systems.*

    ___

    I do not believe this statement is correct, as I was unable to find challenges for future work or a thorough discussion on insights derived from new experiments that can take the ideas proposed in the original work forward. I believe this statement is an over-claim, as the authors only extend the original work and conduct preliminary analysis that goes beyond the original paper (such as measuring entropy vs balance for the attacking fairness metric). While these are good experiments to have, they do not align with the claims made regarding identifying challenges, offering insights, or guiding future efforts in the development of robust + fair AI models. To recommend acceptance, I would ask the authors to either (1) remove this statement completely in the abstract and elsewhere in their paper, or (2) add substantial evidence to the paper backing these claims.
2. The formatting of the paper can be improved. The authors heavily discuss the Appendix figures (Figures 4 and 5) in the main text but defer them to the appendix even though there are additional pages available (12 pages of main content). This experience can be jarring for readers, so I would recommend moving Figures 4 and 5 to the main paper.
3. If the authors have not made some of the figures themselves (ie they are from the original paper), I think they should attribute them in a more clear manner. For instance Figure 3, Figure 6, and Figure 7 should mention they are directly taken from the original paper along with a reference in their captions. Similarly, Section D in the Appendix seems to be directly taken from the original paper (including the footnote). Here I would suggest rewording some of these definitions or directing readers to the original paper if that is sufficient.
4. I have some issues with the Dutch Census + SFD experiments and conclusions drawn in Section 4.2 (additional datasets beyond the original paper). The authors write: _For the attack on the SFD algorithm using the Dutch Census dataset, the attacks do not decrease in fairness scores (balance and entropy) post-attack. ... Through these extended results, it could be said that claims 1 and 2 of the original authors as described in Section 2 are partially supported on datasets that the original authors did not use._

    ___

    This is not quite correct and seems to me has nothing to do with the author's approach but is rather a dataset and clustering model suitability issue. For instance, observe the NMI scores for SFD on Dutch Census (Figure 3). These are very low (between 0.0 and 0.02). These numbers indicate that this clustering model is not very suitable for this dataset and is possibly outputting a random clustering for this dataset. As a result, fairness will be quite high for such a random clustering (this can also be seen in Figure 3 as Balance varies between 0.9 and 1.0). Due to this, it does not make sense to attack this clustering model on this dataset to reduce fairness as it cannot find "good" clustering solutions. Stating that the authors' claims only partially hold because of this is a bit misleading and does not make sense as it is more likely an issue with this model being used on this particular dataset. It is okay to include this result, but I would like the authors to amend this statement to indicate that this is not a particular issue with the attack approach but more likely with the failure of the clustering model being used.

5. There seems to be a typo in Section 4.2: _In Figures 2 and 3 of appendix D_ should read _Figures 2 and 3_ as these figures are in the main paper and not the appendix.
6. I also think because the Yale dataset was removed by the original hosts (as stated in Section 5.2), the authors might be seeing slightly different trends (but as they find, these are still overall consistent with Chhabra et al) owing to the manually downloaded version. I think it would be good to mention this for transparency wherever slight differences are observed in the Yale dataset results.

---

### Review · Reviewer_SDHM · 2024-03-12

**Summary Of Contributions:**

The reproducibility study examines claims made by the paper  "Robust Fair Clustering: A Novel Fairness Attack and Defense Framework". Specifically, the authors confirm that the original claims made by the paper are supported. These claims are

(1) SOTA fair clustering models are susceptible to adversarial attacks that aim to jeopardize the fairness performance.

(2) The proposed Consensus Fair Clustering (CFC) defense mechanism is effective.

The authors confirm the claims by extending the evaluation using additional datasets.

**Audience:**

Yes

**Claims And Evidence:**

Yes

**Requested Changes:**

Please respond to the weaknesses above.

**Strengths And Weaknesses:**

Strengths:
1. The paper is well written and clearly organized.
2. The experiments are comprehensive.


Weaknesses:
1. The paper simply reinforces the original paper’s claim by adding more dataset for evaluation without providing much new theoretical/empirical insights.
2. The meaning of objective function needs further clarification. For example, why does the fairness utility function $\eta$ take $\theta$ and $G_D$ as the parameters? A simple explanation of the intuition of the equation is needed.
3. For section 4.2 "Results beyond original paper",  the new results should be summarized and highlighted for readability.
4. For "Results beyond original paper", do you have speculation/hypothesis why claims 1,2,3 might not hold for additional datasets?

---

### Review · Reviewer_1Q4j · 2024-04-17

**Summary Of Contributions:**

The paper under review is a reproducibility study of the work titled "Robust Fair Clustering: A Novel Fairness Attack and Defense Framework". The primary focus of this study is to validate the original claims about the susceptibility of fair clustering algorithms to adversarial attacks and the efficacy of a proposed defense mechanism called Consensus Fair Clustering (CFC). The reproducibility study confirms the original claims and extends the investigation to additional datasets, providing a robust test of the original paper's conclusions.

**Audience:**

Yes

**Claims And Evidence:**

Yes

**Requested Changes:**

Could you clarify why there is a variation in the pre-attack line in Figure 4?

**Strengths And Weaknesses:**

Strength
The reproducibility study follows a rigorous experimental approach, closely mirroring the original experiments while extending them to additional datasets, thereby enhancing the robustness of the findings.

Weakness
There are potential issues in the experimental results as presented in Figures 1-4. The x-axis in these figures represents the percentage of samples that an attacker can poison. Therefore, at x=0, where no samples are poisoned, the pre-attack and post-attack lines should coincide. However, these lines are observed to be separate at x=0 in most figures, suggesting possible errors in the experimental setup or data handling.

---

> ### Author Response · Authors · 2024-04-17
>
> We want to thank reviewer 1Q4j for their review and constructive feedback. We have implemented all of the feedback and recommendations:
>
> 1. The reviewer pointed out an error in Figures 1 through 4 where not all metrics where equal when 0% of the samples were poisoned. This error was caused by a mistake in the code for plotting, where we labelled the x-axis from 0 through 30%, but entered the data from 12.5 through 30%. We have fixed this and included the datapoints at 0%.
>
> 2. The reviewer also noticed that the pre-attack metrics were varying for the CFC algorithm, while they should be constant. This mistake was caused by the fact that the codebase from the original paper did not apply random seeds for the CFC algorithm, causing variation when running the pre-attack with different percentages of poisoned data. This is also reflected in the Figure 4 of the original paper (Chhabra et al, 2023). We overlooked this problem, and simply re-used their code. However, we have now fixed it by simply taking the mean of all values for the pre-attack for each metric.

---

### Decision · Action_Editor_Dw6m · 2024-05-22

**Recommendation:** Accept with minor revision

**Comment:**

The reviewers are mostly in agreement that the paper can be accepted to TMLR.  One reviewer was more negative and felt that the authors failed to respond to many of the points, but looking over the reviews and responses, I didn't find myself agreeing with that assessment.  (Perhaps some responses themselves could have been more detailed, but I judge the paper changes themselves to be more important.)

For the final version, please perform another careful check over the reviews, as well as the latest post by me (the action editor), and make any final minor changes as needed.

**Audience:**

Attacks and defense methods are a major topic in machine learning.  This paper looks at a fairly specific case of robust fair clustering, but a reproducibility study should be of interest to those working on that topic.

**Claims And Evidence:**

This paper performs a reproducibility study of a specific paper, namely "Robust Fair Clustering: A Novel Fairness Attack and Defense Framework".  The claims of that paper are checked via repeating their experiments as well as performing new ones.  The findings are generally in agreement with the original paper.